# RAGE Mediates Cholesterol Efflux Impairment in Macrophages Caused by Human Advanced Glycated Albumin

**DOI:** 10.3390/ijms21197265

**Published:** 2020-10-01

**Authors:** Adriana Machado-Lima, Raquel López-Díez, Rodrigo Tallada Iborra, Raphael de Souza Pinto, Gurdip Daffu, Xiaoping Shen, Edna Regina Nakandakare, Ubiratan Fabres Machado, Maria Lucia Cardillo Corrêa-Giannella, Ann Marie Schmidt, Marisa Passarelli

**Affiliations:** 1Laboratório de Lípides (LIM 10), Hospital das Clínicas (HCFMUSP), Faculdade de Medicina da Universidade de São Paulo, São Paulo CEP 01246-000, Brazil; prof.adrianalima@usjt.br (A.M.-L.); rodrigo.iborra@saojudas.br (R.T.I.); rspinto@usp.br (R.d.S.P.); enakonda@usp.br (E.R.N.); 2Programa de Pós-Graduação em Ciências do Envelhecimento, Universidade São Judas Tadeu, São Paulo CEP 03166-000, Brazil; 3Department of Medicine, Diabetes Research Program, New York University Langone Health, New York, NY 10016, USA; Raquel.LopezDiez@nyulangone.org (R.L.-D.); gdaffu@gmail.com (G.D.); XiaopingJennifer.Shen@nyumc.org (X.S.); AnnMarie.Schmidt@nyulangone.org (A.M.S.); 4Curso de Biomedicina, Centro Universitário CESMAC, Maceió, Alagoas CEP 57051-160, Brazil; 5Laboratório de Metabolismo e Endocrinologia, Instituto de Ciências Biomédicas da Universidade de São Paulo, São Paulo CEP 05508-000, Brazil; ubiratan@icb.usp.br; 6Laboratório de Carboidratos e Radioimunoensaio (LIM 18), Hospital das Clínicas (HCFMUSP), Faculdade de Medicina da Universidade de São Paulo, São Paulo CEP 01246-000, Brazil; maria.giannella@fm.usp.br; 7Programa de Pós-Graduação em Medicina, Universidade Nove de Julho, São Paulo CEP 01225-000, Brazil

**Keywords:** diabetes mellitus, advanced glycation end products, cholesterol, RAGE, atherosclerosis

## Abstract

We addressed the involvement of the receptor for advanced glycation end products (RAGE) in the impairment of the cellular cholesterol efflux elicited by glycated albumin. Albumin was isolated from type 1 (DM1) and type 2 (DM2) diabetes mellitus (HbA1c > 9%) and non-DM subjects (C). Moreover, albumin was glycated in vitro (AGE-albumin). Macrophages from *Ager* null and wild-type (WT) mice, or THP-1 transfected with siRNA-*AGER,* were treated with C, DM1, DM2, non-glycated or AGE-albumin. The cholesterol efflux was reduced in WT cells exposed to DM1 or DM2 albumin as compared to C, and the intracellular lipid content was increased. These events were not observed in *Ager* null cells, in which the cholesterol efflux and lipid staining were, respectively, higher and lower when compared to WT cells. In WT, *Ager*, *Nox4* and *Nfkb1,* mRNA increased and *Scd1* and *Abcg1* diminished after treatment with DM1 and DM2 albumin. In *Ager* null cells treated with DM-albumin, *Nox4*, *Scd1* and *Nfkb1* were reduced and *Jak2* and *Abcg1* increased. In *AGER*-silenced THP-1, *NOX4* and *SCD1* mRNA were reduced and *JAK2* and *ABCG1* were increased even after treatment with AGE or DM-albumin. RAGE mediates the deleterious effects of AGE-albumin in macrophage cholesterol efflux.

## 1. Introduction

In diabetes mellitus (DM), disturbances in the reverse cholesterol transport (RCT) are related to the pathophysiology of atherosclerosis. The ATP-binding cassette transporters A1 (ABCA1) and G1 (ABCG1) mediate the excess of cholesterol delivered to, respectively, apo A-I and HDL in the first step of the RCT, a pivotal mechanism that helps to maintain lipid homeostasis in the arterial wall favoring cholesterol excretion into bile and feces [1,2].

Advanced glycation end products (AGEs) are prevalent in DM due to hyperglycemia, oxidative stress and inflammation, but can also be absorbed from exogenous sources such as diet and tobacco [3]. AGE adversely affects ABCA1 and G1-mediated cholesterol efflux [4,5] by reducing ABCA1 protein levels in macrophages without changing Abca1 mRNA [6,7]. On the other hand, in a LXR-dependent mechanism, AGEs reduce *ABCG1* gene transcription [8].

The receptor for AGE (RAGE, *AGER* gene) is a multi-ligand receptor of the immunoglobulin superfamily that binds AGE and other inflammatory molecules such as calgranulins and *high-*mobility* group *protein** 1 (HMBG1), also referred to as a pattern recognition receptor [9]. RAGE is highly expressed in atherosclerotic lesions from human and animal models of DM and mediates the deleterious effects of AGE in vasculature [10,11]. Its antagonism by soluble forms that lack transmembrane and signaling domains counteracts the full length receptor signaling, diminishing the development of atherosclerosis in dyslipidemic diabetic mice [12]. In addition, AGER silencing by small interference RNA is able to reduce inflammation and vascular damage [13,14].

We demonstrated that AGE albumin, isolated from both type 1 and 2 DM subjects’ serum, alters the transcription of genes involved in ABCA1 expression and activity in macrophages, such as *Scd1* (Stearoyl-Coenzyme A desaturase 1), *Jak2* (Janus kinase 2) and *Nox4* (NADPH oxidase 4) [15,16], leading to intracellular lipid accumulation. Here, we tested the hypothesis that *AGER* silencing suppresses the reduction in macrophage cholesterol efflux induced by AGE albumin and rescues the gene expression profile. By utilizing bone marrow-derived macrophages (BMDM) from *Ager* null mice and THP-1 cells with *AGER* knockdown, we demonstrate that RAGE mediates the reduction in cholesterol efflux induced by AGE albumin by modulating macrophage gene expression.

## 2. Results

The biochemical parameters of C and DM subjects are shown in Table 1. The body weight, BMI, total cholesterol (TC), HDLc, LDLc and microalbuminuria were similar among groups. The duration of the disease was similar between DM 1 and DM 2 individuals. DM 2 subjects were older than DM 1 and controls, and they presented higher plasma TG levels. Fasting glycemia, HbA1c, fructosamine and total AGE in albumin were similar between DM groups but superior than the C group.

In order to access the role of RAGE in the disturbances of cholesterol efflux elicited by glycated albumin drawn from those patients when compared to albumin from control subjects, bone marrow-derived macrophages (BMDM) from WT and *Ager* null mice were utilized after cholesterol overloading. The percentage of ^3^H-cholesterol efflux mediated by apo A-I was significantly reduced by 71% and 81% in WT BMDM exposed to DM 1 or DM 2 albumin, respectively, as compared to C albumin (Figure 1A). In addition, when compared to C albumin, the HDL_2_-mediated cholesterol efflux in WT BMDMs was significantly reduced by 58% and 49% by DM1 or DM2 albumin, respectively (Figure 1B). These changes were not observed when *Ager* null BMDM were incubated with DM albumins (Figure 1A,B). In addition, the cholesterol efflux mediated by apo A-I and HDL_2_ was higher in *Ager* null mice when compared to WT BMDM cells when both cells were treated with DM albumin (Figure 1, panels A and B). These findings point to a role of the AGE/RAGE axis in the alteration of the ABCA1- and ABCG1-mediated lipid efflux in macrophages.

Similar results were found in WT and *Ager* null BMDM treated with in vitro glycated albumin (AGE albumin): a 67% and 39% reduction in cholesterol efflux mediated, respectively, by apo A-I and HDL_2_ was observed in WT BMDM treated with AGE albumin when compared to cells treated with non-glycated albumin. In the absence of *Ager,* disturbances in the cholesterol efflux were prevented (Figure 1C,D). After treatment with AGE albumin, the cholesterol efflux mediated by HDL_2_ from BMDM isolated from *Ager* null mice was significantly higher when compared to WT cells (Figure 1D).

Oil red O staining was performed to analyze the intracellular lipid accumulation in cells exposed to DM, non-DM, glycated and non-glycated albumins. For that, WT and *Ager* null BMDMs were enriched with acetylated LDL, treated with albumins for 48 h and incubated with apo AI or HDL, following the assessment of lipid staining. The intracellular lipid accumulation increased after treatment with DM1 and DM2 albumin, even after incubation with lipid acceptors, apo AI or HDL (Figure 2A,B). On the other hand, there was a reduction in lipid staining in *Ager* null BMDMs treated with DM1 and DM2 albumin after incubation with apo AI or HDL (Figure 2A,B). Similar results were found in cells treated with AGE albumin; while WT cells showed an increase in intracellular lipid staining by AGE-albumin when compared to non-glycated albumin, this was not observed in *Ager* null BMDMs that were preserved from lipid accumulation (Figure 2C,D). These data are in agreement with the results obtained in cholesterol efflux assays, confirming a role of the AGE/RAGE axis in mediating cholesterol homeostasis disturbances in macrophages.

Gene expression was assessed in BMDM isolated from WT and *Ager* null mice (Figure 3) or in *AGER*-silenced THP-1 cells (Figure 4) treated with albumin samples. Genes were chosen based on our previous findings, which demonstrated their involvement in the ABC-mediated cholesterol efflux in macrophages [15,16]. *Ager* expression was increased in WT BMDMs after treatment with DM 1 or AGE albumin in comparison to, respectively, C and non-glycated albumin (Figure 3A). There was an increase in *Nox4* mRNA expression in WT cells after treatment with DM 1 and DM 2 albumin in comparison to C albumin (Figure 3B). In spite of the presence of DM 1 or DM 2 albumin, a dramatic reduction in *Nox4* expression was observed in *Ager* null cells (Figure 3B). *Jak2* mRNA expression was similar in WT cells treated with C, DM 1 and DM 2 albumin, but after treatment with DM 1 and DM 2 albumin it increased in *Ager* null BMDMs when compared to WT cells (Figure 3C). Nonetheless, *Scd1* mRNA expression was lower in *Ager* null cells than in WT after treatment with DM albumins (Figure 3D). *Nfkb1* increased significantly in WT BMDM treated with DM 1 or DM 2 albumin in comparison to C albumin but was lower in *Ager* null cells than in WT cells (Figure 3E).

RAGE silencing was utilized as another approach to investigate the role of RAGE in intracellular cholesterol homeostasis. The expression of the *AGER* gene in THP-1 cells transfected with siRNA-AGER was reduced by 60% in comparison to scramble siRNA-treated cells (Figure 4A). This agreed with the fact that the amount of RAGE assessed by immunoblot decreased by 77% in *AGER*-silenced THP-1 cells in comparison to scramble siRNA-cells (Figure 4B).

In *AGER*-silenced THP-1 cells, there was a reduction in the *NOX4* mRNA expression after treatment with AGE albumin when compared to cells transfected with scramble siRNA (Figure 4C). No statistical differences were obtained regarding DM 1 and DM 2 albumin. *JAK2* increased in siRNA-*AGER* cells treated with AGE or DM 2 albumin when compared to scramble siRNA cells (Figure 4D). In scramble siRNA cells, *SCD1* increased via AGE and DM 2 albumin when compared to their respective experimental controls (non-glycated and C albumin) and decreased after treatment with DM 1 albumin. In siRNA-*AGER* cells, the expression of *SCD1* decreased under all experimental conditions when compared to scramble siRNA cells (Figure 4E).

The expressions of the cholesterol transporters ABCA1 and ABCG1 were also studied in both BMDMs (Figure 5A,C) and THP-1 cells (Figure 5B,D). The *Abca1* levels increased via DM 1 albumin in *Ager* null when compared to WT BMDMs. In comparison to C albumin, only DM 2 albumin increased *Abca1* in WT cells, although the mRNA levels were reduced in *Ager* null when compared to WT BMDMs (Figure 5A). In THP-1 cells transfected with scramble siRNA or siRNA-*AGER,* the *ABCA1* mRNA expression was not modified after treatment with DM 1 and DM 2 when compared with C albumin (Figure 5B). A reduction in *Abcg1* was observed in WT BMDMs treated with DM 1 and DM 2 albumin when compared with C albumin. However, there was an increase in *Abcg1* expression in *Ager* null cells exposed to DM 1 albumin when compared to C albumin (Figure 5C). In THP-1 cells transfected with siRNA-*AGER* and exposed to DM 1 or DM 2 albumin, we found an increase in *ABCG1* when compared to cells transfected with scramble siRNA. In addition, DM 2 albumin increased when compared to C albumin in *AGER*-silenced cells (Figure 5D).

## 3. Discussion

Glycated albumin is an important clinical marker of glycemic control and independently predicts long-term outcomes in DM [17]. AGE albumin plays a potential atherogenic role, particularly via its deleterious effects in macrophage reverse cholesterol transport [16]. Considering the involvement of the AGE/RAGE axis in the development of inflammation and vascular damage in DM, we addressed how RAGE is involved in the impairment of apo A-I and HDL-mediated cholesterol efflux elicited by human AGE albumin in macrophages.

By utilizing two different experimental approaches to abrogate RAGE signaling (*Ager* null mouse macrophages and RAGE-silenced THP-1 cells), we demonstrated that: (1) the reduction in cholesterol efflux to apo A-I and HDL_2_ elicited by AGE requires RAGE and that (2) RAGE absence abolishes the effect of AGE albumin, normalizes the expression of many genes involved in cholesterol efflux and prevents intracellular lipid accumulation.

Together with oxidative stress and epigenetics, AGEs are important mediators of metabolic memory. They mediate the intracellular glycation of the mitochondrial respiratory chain proteins, leading to the excessive production of reactive oxygen species, NF-KB activation and increased expression of adhesion molecules and cytokines. In addition to altering the transcription of several genes, AGE induces RAGE expression, creating a vicious cycle in the pathophysiology of vascular damage [18,19].

The in vivo modification of albumin, analyzed in the present study, which occurs in poorly controlled DM individuals, is reflected by the high amount of AGE in DM 1 and DM 2 albumin as compared to C albumin. We previously demonstrated that glycation that takes place in vivo in DM subjects elicits similar alterations in macrophage cholesterol homeostasis when compared to in vitro-produced AGE albumin [6,15,16,20]. Apart from changes in its message level, AGE albumin induces intracellular lipid accumulation as a consequence of the reduction in ABCA1 protein [15,16]. The accumulation of toxic cholesterol derivatives such as 7-ketocholesterol has been described as being associated to the enhanced inflammation, as well as the oxidative and endoplasmic reticulum stress, in macrophages [6,7,21,22].

In the present work, the absence of RAGE prevented alterations in the macrophage cholesterol efflux induced by both sources of glycated albumin, isolated from poorly controlled DM subjects and produced in vitro. This result agrees with recent observations from our group [23] demonstrating that a 54% RAGE knockdown in THP-1 cells prevented the reduction in ABCA-1 elicited by AGE-albumin. Those results likely point to a role of RAGE in mediating the activation of the ubiquitin-proteasome and lysosomal-related degradation pathways that are responsible for the intracellular degradation of ABCA-1 protein elicited by AGE.

In addition, Daffu et al. (2015) demonstrated that *AGER* knockdown rescued *ABCG1* expression and HDL-mediated cholesterol efflux in cells treated with carboxymethyllysine (a specific RAGE ligand) [5]. Altogether, these results point to a major role of RAGE in mediating the AGE effects in cholesterol homeostasis, although they do not exclude the participation of other scavenger receptors and toll-like receptors that may bind AGE.

Our findings agree with previous clinical observations in DM subjects, where the circulating levels of AGE and soluble forms of RAGE were associated with the progression of cardiovascular disease. In DM 2 individuals with stable coronary artery disease, the levels of glycated albumin and the splice variant endogenous secretory RAGE (esRAGE) were independent predictors of primary and secondary endpoints [24]. In mononuclear cells, the expression of esRAGE decreased in pre-diabetes and type 2 DM subjects when compared to controls and was, together with HbA1c, a determinant of the intima-media thickness [25]. In addition, in type 1 DM soluble RAGE (sRAGE), the levels were inversely related to inflammation in a five-year follow-up study [26].

Gene expression was assessed in BMDMs isolated from *Ager* null mice or in *AGER*-silenced THP-1 cells treated with C or DM albumin samples. The *Ager*, *Nox4* and *Nfkb1* expression increased in WT BMDMs after treatment with DM albumin. The deletion of *Ager* decreased *Nox4* after DM 1 or DM 2 albumin treatment, indicating a role of RAGE silencing in the modulation of oxidative stress that is deleterious to cholesterol exportation via ABCA1. *Jak2* mRNA expression was increased in *Ager* null BMDMs when compared to WT cells after treatment with DM 1 and DM 2 albumin. JAK-2 is postulated to be a mediator of the apo A-I and ABCA-1 interaction, a requirement for free cholesterol exportation that leads to HDL assembly [27]. RAGE induces JAK-2/STAT (signal transducer and activator of transcription) activation [28,29], and its silencing is able to prevent this intracellular signaling [14].

In *Ager* null cells, when compared to WT cells, there was a reduction in *Nfkb1* mRNA expression even after treatment with albumin isolated from DM subjects. The AGE/RAGE interaction mediates oxidative stress generation [30] that evokes NF-κB activation [31], thereby increasing the chronic inflammatory and vasoconstrictor response related to long-term complications of DM.

There was a reduction in *Scd1* in *Ager* null cells and in siRNA-*AGER* THP-1 cells treated with DM 1 and DM 2, which may be beneficial considering the negative modulation of *ABCA1* mRNA levels by SCD-1 [32,33]. Nonetheless, the role of SCD-1 in atherogenesis is still controversial, since a low level of SCD-1 activity in macrophages triggers endoplasmic reticulum stress that leads to a reduction in the ABCA1 protein level [22].

The expression of *Abca1* was increased by DM 1 albumin in *Ager* null BMDMs when compared to WT cells. DM 2 albumin enhanced *Abca1* mRNA in WT cells when compared to C albumin, which may be the reason for the difference observed between WT and *Ager* null cells treated with DM 2 albumin. On the other hand, no differences in *ABCA1* expression were observed in scramble siRNA- or siRNA-*AGER*-transfected THP-1 cells treated with DM albumin when compared to C albumin. It is worth noting that the final content of ABCA1 protein in cells is mainly dictated by posttranslational mechanisms represented by protease-mediated degradation, ubiquitin-proteasome and lysosomal degradation [34,35]. Thus, the elevation in *ABCA1* mRNA levels that we observed may not account for the final protein level, which is reduced by AGE albumin [7,16], as described by the increased ABCA-1 ubiquitination and degradation [23].

A reduction in *Abcg1* mRNA expression was observed in WT BMDMs treated with DM 1 and DM 2 albumin. *Abcg1* mRNA increased when *Ager* null cells or THP-1 cells transfected with siRNA-*AGER* were exposed to DM albumin. Daffu et al. (2015) found a reduction in *Abca1* and *Abcg1* mRNA in RAGE-expressing diabetic BMDMs when compared to *Ager* null diabetic BMDMs [5]. In their paper, they demonstrated that the AGE/RAGE axis downregulates the luciferase activity in the *ABCG1* promoter and decreases the cholesterol efflux to HDL. Although changes in gene expression reported here were not confirmed by evaluating the protein expression, the prevention of disturbances in the cholesterol exportation to apo A-I and HDL, as well as in intracellular lipid accumulation, observed in the absence of RAGE reinforce the role of AGE-RAGE signaling in the deleterious effects of AGEs on lipid homeostasis in macrophages. In addition, the use of in vitro-glycated albumin confirms the specific role of AGEs, generated both in vitro and in vivo, in the diabetes mellitus milieu. The recovery of cellular functionality that favored cholesterol efflux and reduced intracellular lipid accumulation seems to represent a cellular whole integration (gene expression and protein levels/functions) that is attained by the absence of RAGE signaling.

In another investigation, we found that serum albumin drawn from an animal model of uremia also renders macrophages vulnerable to endoplasmic reticulum stress and disturbs reverse cholesterol transport by impairing the ABCA1 expression and activity [36]. Interestingly, glycated albumin is enhanced in those animals’ serum and acts similarly to glycated albumin drawn from DM subjects. RAGE is overexpressed in the uremic aortic wall, and *Ager* null animals are protected from the uremia-induced acceleration of atherosclerosis [37]. Thus, RAGE inhibition may contribute to abrogating the deleterious effects of AGE albumin in a range of metabolic conditions where carbonyl stress prevails.

In conclusion, cholesterol efflux impairment and intracellular lipid accumulation induced by human advanced glycated albumin is due to alterations in macrophage gene expression mediated by the AGE/RAGE axis. Strategies to block RAGE signaling might be useful in preventing derangements in macrophage reverse cholesterol transport and atherosclerosis induced by AGE.

## 4. Materials and Methods

Control individuals (*n* = 7) were selected at the Faculdade de Medicina da Universidade de São Paulo. Type 1 (*n* = 7) and type 2 DM (*n* = 9) subjects with HbA1c > 9% were selected at the Hospital das Clínicas da Faculdade de Medicina da Universidade de São Paulo (HCFMUSP). All participants signed an informed written consent form previously approved by The Ethical Committee for Human Research Protocols of HCFMUSP (CAPPesq protocol #195.421; 21 January 2013). None of the participants had chronic diseases other than DM, and subjects diagnosed with microalbuminuria, renal disease or alcohol abuse were excluded. One individual in the control group was a current smoker, two were on enalapril and one on L-thyroxine treatment. In the DM 1 group, all subjects were on insulin treatment and none were smokers. In the DM 2 group, six were on insulin, nine on statins, six on sulfonylureas, nine on metformin, two on acarbose, one on pioglitazone, two on fibrate, eight on antihypertensives and six on acetylsalicylic acid. One subject was a current smoker.

Plasma fructosamine, glucose, triglycerides, total cholesterol, HDL cholesterol and serum albumin were determined after overnight fasting by enzymatic techniques. HbA1c was determined by high performance liquid chromatography. Urinary albumin was quantified by colorimetric analysis.

### 4.1. Isolation and Purification of Serum Albumin

Serum albumin was isolated from control and DM individuals by fast protein liquid chromatography using a HiTrap^TM^ Blue (GE Healthcare, Uppsala, Sweden) affinity column, following purification by alcoholic extraction, as previously described by Machado-Lima et al., 2015 [16]. The samples’ integrity was assessed by electrophoresis in comparison with a pure commercially available human albumin serum (Sigma-Aldrich, Steinheim, Germany). The amount of endotoxin in the albumin samples was less than 50 pg of endotoxin/mL according to the Limulus Amebocyte Lysate (LAL) (Cape Cod, Falmouth, MA, USA), and no cell toxicity was observed.

### 4.2. In Vitro Advanced Glycation of Human Albumin

Advanced glycation was induced in human fatty acid-free albumin (Sigma-Aldrich, Steinheim, Germany) by incubating with 10 mM glycolaldehyde (Sigma Chemical Co., St. Louis, MO, USA) for four days under sterile conditions in a shaker bath at 37 °C in the dark, under N_2_ atmosphere. Control non-glycated albumin was incubated with phosphate buffered saline only. The samples were dialyzed and, after sterilization, were frozen at −80 °C until experiments.

### 4.3. Total AGE Measurement

The content of total AGE (mU AGE/mg of albumin) in albumin isolated from control individuals and DM 1 and DM2 subjects was quantified by an immunoenzymatic method from LAMIDER SA (ELISA kit for the detection and quantification of AGEs—Mexico DF, México).

### 4.4. Isolation of Plasma Lipoprotein

Plasma preparative ultracentrifugation was performed in order to isolate LDL (d = 1.019–1.063 g/mL) and HDL_2_ (d = 1.063–1.125 g/mL). LDL was acetylated with acetic anhydride, as previously described [38], and protein was measured by the Lowry technique [39]. Purified apo A-I was purchased from Biomedical Technologies (Tewksbury, MA, USA).

### 4.5. L929 Cell Culture

L929 cells (ATCC, American Tissue Culture Collection) were cultured in low glucose DMEM supplemented with 10% heat-inactivated fetal calf serum and 1% penicillin/streptomycin for seven days as a source of colony-stimulating factor-1 (CSF-1; required for bone marrow cells’ differentiation into macrophages) [40]. The medium was removed and stored at −20 °C (first week medium). Confluent monolayers were cultured with fresh medium for seven more days to generate a second batch of conditioned medium (second week medium).

### 4.6. Isolation of Mouse Bone Marrow Cells

Mouse bone marrow cells were obtained from C57BL/6 wild-type (WT) and *Ager* null mice. Male and female homozygous Ager^−/−^ mice, backcrossed > 12 generations into a C57BL6/J background, were bred in our laboratory [41]. Male and female C57BL6/J (WT) mice were purchased from Jackson Laboratories (Bar Harbor, ME, USA). Animal studies were carried out with the approval of the Institutional Animal Care and Use Committee of New York University. Briefly, tissues from legs were removed with scissors and dissected away from the body. All remaining tissue from the femurs and tibias were cleaned and separated at the knee joint to avoid contamination. The end of each bone was cut off and, using a needle size of 26 and ½ and a 20 mL syringe, was filled with bone marrow medium (low-glucose DMEM with 0.8% penicillin/streptomycin, 10% heat-inactivated fetal calf serum and 10% L929-cell conditioned medium [40]—half from the first week and half from the second week). The bone marrow from both ends of the bones was expelled with a jet of medium directed into a 50 mL screw top tube. Using a needle size 18 and ½ attached to a 20 mL syringe, the marrow was gently aspirated and expelled until the cell aggregates were broken up. Cells were centrifuged for 6 min at 1000 rpm at room temperature. Cells were resuspended in bone marrow medium and were dispensed into culture dishes. Incubation was done for five days at 37 °C under a 5% (*v*/*v*) CO_2_. On day 5, the conditioned medium was changed for a new bone marrow medium. On day 6, the growth medium was completely changed to normal medium (low-glucose DMEM + 1% penicillin/streptomycin + 10% heat-inactivated fetal calf serum).

### 4.7. Cholesterol Efflux Assay

Bone marrow-derived macrophages (BMDM) from WT and *Ager* null mice were incubated with acetylated LDL (50 µg/mL) and 5 µCi/mL of ^3^H-cholesterol for 24 h. After two washes with PBS containing fatty acid free albumin (FAFA), cells were maintained for 48 h in DMEM containing 1% penicillin/streptomycin supplemented with 1 mg/mL of albumin from control subjects, DM 1 or DM 2 subjects. In another set of experiments, cells were incubated with non-glycated or with in vitro glycated albumin (AGE albumin). Macrophages were then incubated with 50 µg/mL HDL_2_ or 30 µg/mL apo A-I for 6 h to determine the ^3^H-cholesterol efflux, as previously described [4].

### 4.8. Intracellular Lipid Staining

Bone marrow-derived macrophages (BMDMs) from WT and *Ager* null mice were seeded in chamber slides (Lab Tek) and were then enriched with acetylated LDL (50 µg/mL) for 24 h. After two washes with PBS containing fatty acid free albumin (FAFA), cells were maintained for 48 h in DMEM containing 1% penicillin/streptomycin supplemented with 1 mg/mL of albumin from control, DM 1 or DM 2 subjects. In another set of experiments, cells were incubated with non-glycated or in vitro glycated albumin (AGE albumin). Then, macrophages were incubated with 50 µg/mL HDL or 30 µg/mL apo AI for 6 h. Cells were washed with PBS and fixed with formalin solution (10% in PBS) for 10 min at room temperature. Cells were washed with PBS and 60% isopropanol. Then, cells were stained with Oil Red O working solution: 30 mL of the stock stain in 220 mL of distilled water (Oil Red O stock stain: 0.5 g of Oil Red O—Sigma-Aldrich—dissolved in 100 mL of isopropanol) for 25 min. Cells were washed with 60% isopropanol and were stained with hematoxilin (stain nucleo core) for 1 min. Cells were washed with distilled water, and aqueous mounting medium was used to put the coverslips. Cells were observed under an optical microscope (Sony CCD Camera/Olympus Microscope BX-51). Oil Red O staining was quantified by a single blinded investigator using Image Pro Plus Media Cybernetics software (Bethesda, Rockwell, MD, USA). Data were expressed as the stained area detected by the software Image Pro Plus Media Cybernetics software (Bethesda, Rockwell, MD, USA).

### 4.9. AGER siRNA

The human monocytic leukemia cell line, THP-1 (ATCC, American Tissue Culture Collection) was seeded in 6-well plates and transfected with small interfering RNA (siRNA) duplexes against *AGER*. The siRNA duplexes against *AGER* (AM16706, Ambion) were electroporated into THP-1 cells using serum-free medium (Opti-MEM^®^ Reduced Serum Medium, Gibco, Paisley, Scotland, UK) without antibiotic supplements, using Lipofectamine^®^ 2000 Transfection Reagent (Invitrogen, Thermo Fisher Scientific Inc., Waltham, MA, USA), according to the manufacturer’s protocol. To control for the off-target effects of siRNA, separated wells of THP-1 cells were electroporated with scramble siRNA (AM4635, Ambion) as negative controls. Cells were incubated under these conditions for 96 h. After that, cells were maintained for 48 h in DMEM supplemented with 1 mg/mL of albumin from control, DM 1 or DM 2 subjects. In another set of experiments, cells were incubated with non-glycated or with AGE albumin. The total RNA was extracted from THP-1 cells using the RNeasy Mini Kit (Qiagen, Hilden, Germany).

### 4.10. Real-Time Quantitative PCR

The total RNA (0.5 µg) was processed directly to cDNA using the iScript™ cDNA Synthesis Kit (Bio-Rad Laboratories, Inc., Foster City, CA, USA), according to the manufacturer’s protocol. Real-time quantitative PCR was performed using TaqMan Universal PCR Master Mix (Applied Biosystems, Foster City, CA, USA). TaqMan Gene Expression Assays were used in the Step One Plus Real Time PCR System (Applied Biosystems, Foster City, CA, USA) (Table 2).

The relative expression of each gene was measured with respect to the expression of the housekeeping gene *IPO8*. The relative quantification of the gene expression was performed with StepOne Software 2.0 (Applied Biosystems, Foster City, CA, USA) using the comparative cycle threshold (Ct) (2^−ΔΔCt^) method [42,43].

### 4.11. Statistical Analysis

Statistical analyses were performed using GraphPad Prism 8.0 software (GraphPad Prism, Inc., San Diego, CA, USA). The Shapiro–Wilk normality test was applied, and the one-way ANOVA with Dunnett´s posttest or Student´s t test were utilized to compare results (mean ± SD). A *p*-value < 0.05 was considered statistically significant.

## Figures and Tables

**Figure 1 ijms-21-07265-f001:**
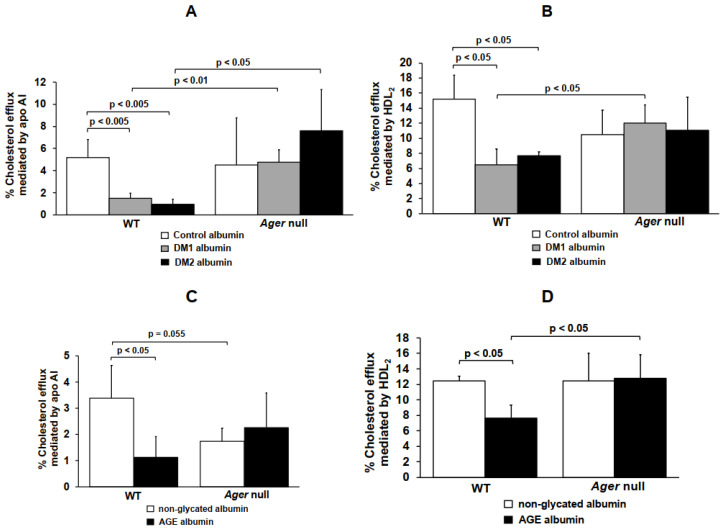
Cholesterol efflux mediated by apo AI and HDL2 in macrophages treated with non-glycated, AGE, control, DM 1 or DM 2 albumin. Bone marrow-derived macrophages (BMDMs) from C57BL/6 WT (n = 4) mice and Ager null (n = 4) mice were loaded with acetylated LDL (50 µg/mL) and 3H-cholesterol for 24 h. Cells were treated for 48 h with (A and B) control, DM 1 or DM 2 albumin or with (C and D) non-glycated or AGE albumin (1 mg/mL in DMEM). Apo A-I (30 µg/mL) or HDL2 ((50 µg/mL) were utilized as 3H-cholesterol acceptors in 6-h incubations. One-way ANOVA—Dunnett´s posttest; mean ± SD.

**Figure 2 ijms-21-07265-f002:**
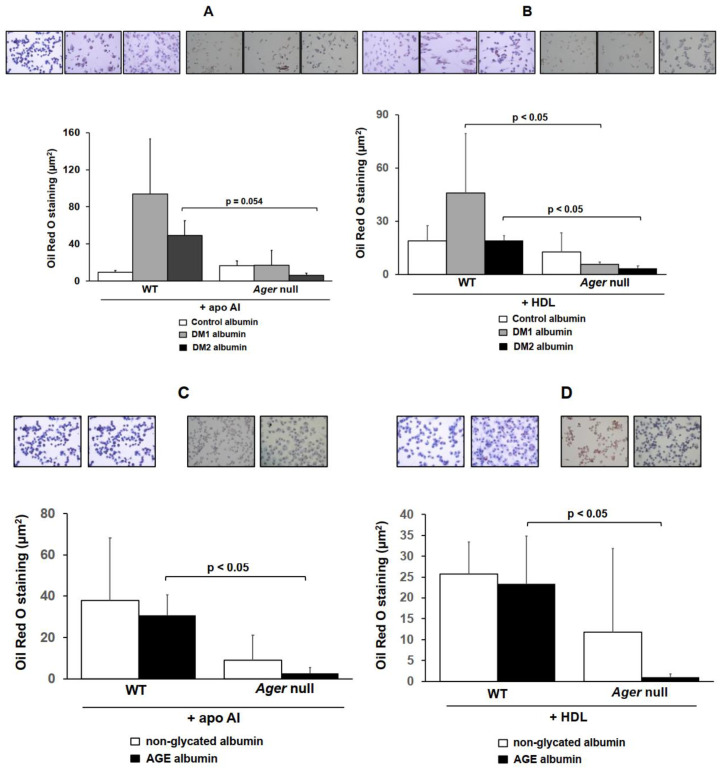
Intracellular lipid staining in macrophages treated with non-glycated, AGE, control, DM 1 or DM 2 albumin. Bone marrow-derived macrophages (BMDMs) from C57BL/6 Wild-type (WT, *n* = 3) and RAGE knockout (RAGE-KO, *n* = 3) mice were loaded with acetylated LDL (50 µg/mL) for 24 h. Cells were treated for 48 h with control, DM 1 or DM 2 albumin (**A**,**B**) or with non-glycated or AGE albumin (**C**,**D**) (1 mg/mL in DMEM) alone or in the presence of apo AI (30 µg/mL) or HDL (50 µg/mL) for 6 h to determine the Oil Red O staining. Representative images (400x magnification). Student’s t test; mean ± SD.

**Figure 3 ijms-21-07265-f003:**
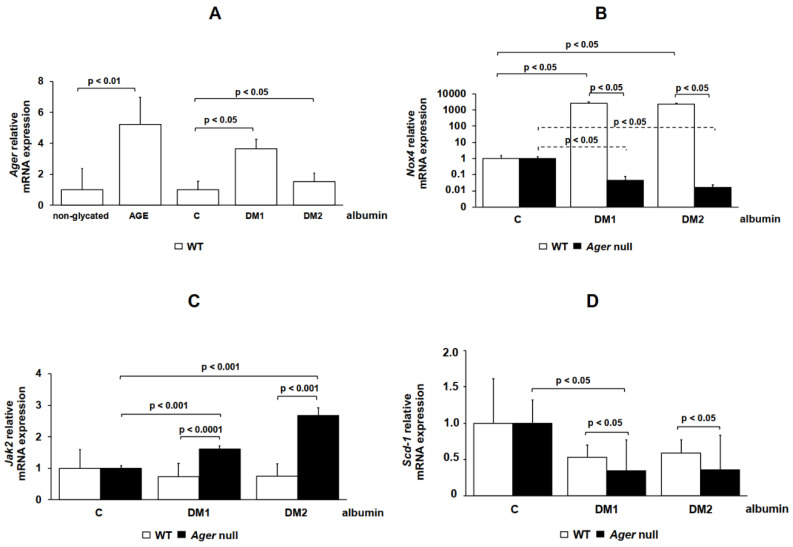
Gene expression in WT and *Ager* null macrophages treated with control, DM 1 or DM 2 albumin. Bone marrow-derived macrophages (BMDMs) from C57BL/6 WT (*n* = 3) and *Ager* null (*n* = 3) mice were maintained for 48 h in DMEM (1 mg/mL) with control, DM 1 or DM 2. RT qPCR was performed for (**A**) *Ager*, (**B**) *Nox4*, (**C**) *Jak2*, (**D**) *Scd1* and (**E**) *Nfkb1* using TaqMan Universal PCR Master Mix (Applied Biosystems). *IPO8* rRNA (Applied Biosystems) was used as an endogenous reference gene. One-way ANOVA—Dunnett’s posttest; mean ± SD.

**Figure 4 ijms-21-07265-f004:**
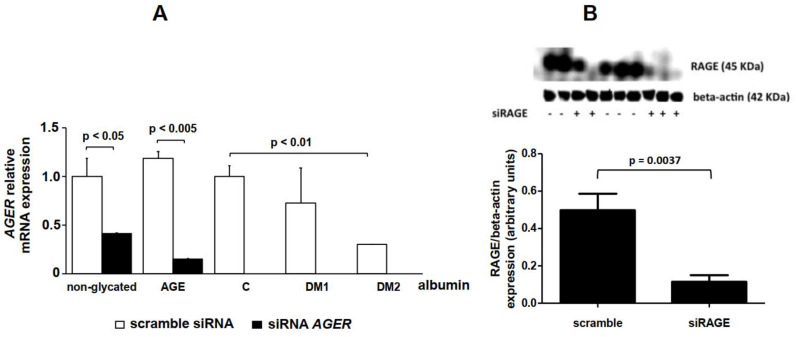
Gene expression in THP-1 cells transfected with scramble siRNA and siRNA-*AGER.* THP-1 cells (*n* = 4) were transfected with small interfering RNA (siRNA) duplexes against *AGER* (AM16706, Ambion, Austin, TX, USA). Separated wells of THP-1 cells were electroporated with a scramble siRNA (AM4635, Ambion) as a negative control. After 96 h, cells were maintained for 48 h in DMEM containing 1 mg/mL of non-glycated, AGE, control, DM or DM 2 albumin. RT qPCR was performed for (**A**) *AGER*, (**C**) *NOX4*, (**D**) *JAK2* and (**E**) *SCD1* using TaqMan Universal PCR Master Mix (Applied Biosystems). *IPO8* rRNA (Applied Biosystems) was used as an endogenous reference gene. Protein levels of RAGE by Western blotting (**B**). One-way ANOVA—Dunnett’s posttest; mean ± SD.

**Figure 5 ijms-21-07265-f005:**
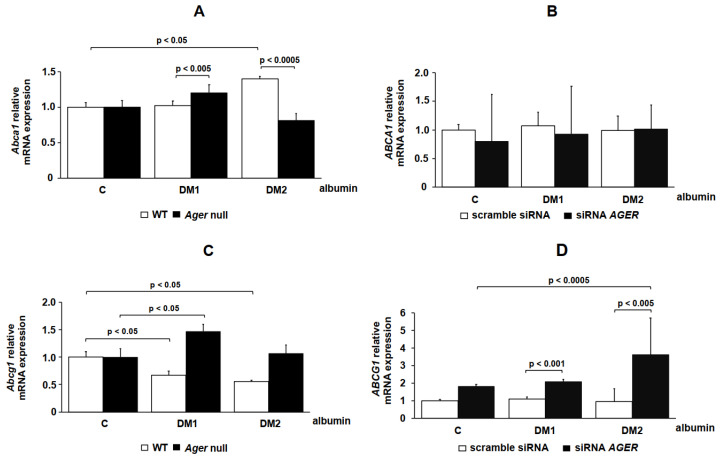
*ABCA1* and *ABCG1* gene expression in cells treated with C, DM 1 and DM 2 albumin. (**A**,**C**) Bone marrow-derived macrophages (BMDMs; *n* = 3) from C57BL/6 WT and *Ager* null mice or (**B**,**D**) THP-1 cells (*n* = 4) transfected with scramble siRNA and siRNA-*AGER* were treated with C, DM 1 or DM 2 albumin (1 mg/mL), as described in the Material and Methods. RT qPCR was performed using TaqMan Universal PCR Master Mix (Applied Biosystems). *IPO8* rRNA (Applied Biosystems) was used as an endogenous reference gene. One-way ANOVA—Dunnett’s posttest; mean ± SD.

**Table 1 ijms-21-07265-t001:** Clinical and biochemical data from control, DM 1 and DM 2 subjects.

	Control (*n* = 7)	DM 1 (*n* = 7)	DM 2 (*n* = 9)
Female/Male	5/2	5/2	6/3
Age (years)	28 ± 2	26 ± 3	63 ± 3 **
Weight (kg)	74.9 ± 8.5	64.3 ± 4	73.9 ± 4.2
BMI (kg/m^2^)	25.5 ± 1.8	23 ± 1.2	30.1 ± 1.7
Duration of DM (years)	-	14 ± 2	14 ± 2
Total cholesterol (mg/dL)	170 ± 8	156 ± 7	187 ± 8
HDL-c (mg/dL)	63 ± 7	55 ± 4	55 ± 10
LDL-c (mg/dL)	87 ± 7	85 ± 8	98 ± 5
Triglycerides (mg/dL)	103 ± 16	80 ± 14	197 ± 37 *
Urinary albumin (mg/dL)	5.6 ± 1.2	13.5 ± 3.5	10.9 ± 2.3
Glucose (mg/dL)	80 ± 2	170 ± 39 *	192 ± 25 *
HbA1c (%)	5.3 ± 0.1	9.6 ± 0.4 **	10.2 ± 0.4 **
Fructosamine (μmol/L)	245 ± 11.7	433 ± 34 **	351 ± 10.7 **
Total AGE (mU AGE/mg of albumin)	12.7 ± 1.5	38.5 ± 1.4 **	35.6 ± 0.6 **

* *p*-value < 0.05; ** *p*-value < 0.0001 compared to Control subjects (One-way ANOVA - Dunnett´s post test; mean ± SE); Statistical analyses were performed using GraphPad Prism 5.0 software (GraphPad Prism, Inc., San Diego, CA, USA). Populational reference values for Total cholesterol (<190 mg/dL), HDL-c (>40 mg/dL), LDL-c (<130 mg/dL), Triglycerides (<150 mg/dL), Glucose (≤99 mg/dL), HbA1C (≤6.5%), Fructosamine (205 to 285 µmol/L), and urinary albumin (30 a 50 g/L)

**Table 2 ijms-21-07265-t002:** Primers utilized-TaqMan Gene Expression Assays (Step One Plus Real Time PCR).

GENE	Human ID	Mouse ID
*AGER* (RAGE)	Hs00153957_m1 (FAM 75 μL 20×)	Mm01134790_g1 (FAM 75 μL 20×)
*NOX4* (NADPH oxidase 4)	Hs00418356_m1 (FAM 75 μL 20×)	Mm00479246_m1 (FAM 75 μL 20×)
*JAK2* (Janus kinase 2)	Hs00234567_m1 (FAM 75 μL 20×)	Mm01208489_m1 (FAM 75 μL 20×)
*SCD1* (Stearoyl-CoA desaturase-1)	Hs01682761_m1 (FAM 75 μL 20×)	Mm00772290_m1 (FAM 75 μL 20×)
*NFKB1* (NF-kappaB)	Hs00765730_m1 (FAM 75 μL 20×)	Mm00476361_m1 (FAM 75 μL 20×)
*ABCA1* (ATP binding cassette subfamily A member 1)	Hs01059118_m1 (FAM 75 μL 20×)	Mm00442646_m1 (FAM 75 μL 20×)
*ABCG1* (ATP binding cassette subfamily G member 1)	Hs00245154_m1 (FAM 75 μL 20×)	Mm00437390_m1 (FAM 75 μL 20×)
*IPO8* (Importin 8)	Hs00183533_m1 (VIC 75 μL 20×)	Mm01255158_m1 (VIC 75 μL 20×)

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
