# Peer review of "RAGE Mediates Cholesterol Efflux Impairment in Macrophages Caused by Human Advanced Glycated Albumin"

_ijms, 2020, doi:10.3390/ijms21197265_

Round 1

Reviewer 1 Report

The authors provided a point-by-point response letter. However, in most cases, they do not properly addressed the reviewer's concerns.

1) "The authors do believe that the cholesterol efflux measurement in THP-1
silenced cells will be similar to that in ager null cells". This may be probably true, but it remains a pure speculation that the authors needed to address to convince about the reproducibility of the effects also in other experimental models.

2) "The authors fully agree with the reviewer´s comment. One of the reasons for the delay in responding to the reviewers was due to numerous attempts and methodological failures to confirm by Western blots our gene expression
results." The authors should explain why Western blot results were difficult to obtain, especially because, as already stated, the function is exerted by the protein and not by the mRNA which is exclusively informative for evaluating prospective molecular mechanisms associated to protein changes.

3) "The ABCA-1 protein expression has already been reported in previous publications from our group in very similar experimental conditions utilizing RAGE silencing. This was mentioned in the discussion session (Iborra et al, 2018)". The support of literature data is not enough to provide convincing hints about ABCA1 expression levels in this experimental context. Indeed, as far as I can see in previous works, the authors never assessed ABCA1 protein expression in Ager-null mice. It is mandatory to evaluate ABCA1 protein levels in cells derived from Ager-null mice, compared with cells obtained from WT mice, treated or not with DM1 or DM2 albumin.

4) "The authors included experiments dealing with intracellular lipid staining by Oil Red O as shown in Figure 2". Representative images for Oil Red staining should be included together with the densitometric analysis.

5) "The authors expressed the results as mean ± SEM and this was properly informed in all figure legends. This was utilized to ameliorate graphical presentation but did not interfere or even mask statistical significances that were attained by utilizing adequate statistical analysis." SEM and SD have distinct and specific meanings, they are not interchangeable just to ameliorate the graphical representation. In this context, SD represents a more accurate manner to show data, since it show (definitely more precisely than SEM) the variability of the population. This is particularly relevant when the sample size is so minimal (n=3 in some cases).

6) Bars in figure 1 and 2 are cutted

Author Response

Reviewer 1

The authors provided a point-by-point response letter. However, in most cases, they do not properly addressed the reviewer's concerns.

1) "The authors do believe that the cholesterol efflux measurement in THP-1
silenced cells will be similar to that in ager null cells". This may be probably true, but it remains a pure speculation that the authors needed to address to convince about the reproducibility of the effects also in other experimental models.

2) "The authors fully agree with the reviewer´s comment. One of the reasons for the delay in responding to the reviewers was due to numerous attempts and methodological failures to confirm by Western blots our gene expression
results." The authors should explain why Western blot results were difficult to obtain, especially because, as already stated, the function is exerted by the protein and not by the mRNA which is exclusively informative for evaluating prospective molecular mechanisms associated to protein changes.

3) "The ABCA-1 protein expression has already been reported in previous publications from our group in very similar experimental conditions utilizing RAGE silencing. This was mentioned in the discussion session (Iborra et al, 2018)". The support of literature data is not enough to provide convincing hints about ABCA1 expression levels in this experimental context. Indeed, as far as I can see in previous works, the authors never assessed ABCA1 protein expression in Ager-null mice. It is mandatory to evaluate ABCA1 protein levels in cells derived from Ager-null mice, compared with cells obtained from WT mice, treated or not with DM1 or DM2 albumin.

The authors recognize that adding experiments with different cell lines to the manuscript is important to reinforce biological mechanisms independently of cell specificity. This was the reason we utilized two different approaches to impair RAGE signaling: macrophages obtained from RAGE KO mice and Ager silencing in THP-1 cells. Each cell line utilized presents different characteristics that favored the specific experiments that were performed allowing very confident results. In this sense, the authors consider that experiments are complementary even though obtained in distinct cells. Besides, they agree with our previously published observations that showed impairment in cholesterol homeostasis induced by AGEs that is related to the reduction in ABCA-1 expression in different cell lines such as mouse peritoneal macrophages, THP-1 cells, bone-marrow-derived macrophages, and J744 cells. Those previous results made us confident regarding the results described in the present investigation. The reviewer´s comment “the function is exerted by the protein and not by the mRNA which is exclusively informative for evaluating prospective molecular mechanisms associated with protein changes” is well known but changes in protein functionality may overcome their expression level. For that reason, the authors reaffirm that cellular functionality (specifically related to lipid efflux) inferred by measuring cholesterol efflux and lipid accumulation is even more informative than proteins as a final indicator of the lipid homeostasis modulation in cells where RAGE signaling was impaired (namely macrophages from RAGE KO mice or Ager knockdown THP-1 cells).

Being honest, the authors tried many times to perform immunoblot assays to analyze protein expression but the bands were not well visualized due to technical problems with the secondary antibodies. Unfortunately, nowadays it is difficult to import new reagents and students are on quarantine due to the SarsCov pandemic which makes access to laboratories involved in the study prohibited.   We kindly ask the reviewer the consideration of the functional aspects that ultimately reflects the alteration in gene and protein expression (although the latter was not direct assessed).

4) "The authors included experiments dealing with intracellular lipid staining by Oil Red O as shown in Figure 2". Representative images for Oil Red staining should be included together with the densitometric analysis.

As suggested, the authors included representative images for oil Red O staining

5) "The authors expressed the results as mean ± SEM and this was properly informed in all figure legends. This was utilized to ameliorate graphical presentation but did not interfere or even mask statistical significances that were attained by utilizing adequate statistical analysis." SEM and SD have distinct and specific meanings, they are not interchangeable just to ameliorate the graphical representation. In this context, SD represents a more accurate manner to show data, since it show (definitely more precisely than SEM) the variability of the population. This is particularly relevant when the sample size is so minimal (n=3 in some cases).

The authors are thankful for the explanation and as suggested, figures were changed presenting include SD instead of SEM.

6) Bars in figure 1 and 2 are cutted

The figures were properly adjusted

Reviewer 2 Report

This is a manuscript that is very nicely written and discussed. The introduction is excellent with just a few minor flaws (such as different fonts, for example on line 48). 

In the result section, I suggest a better transition when going from the human subject to the mice. It feels a bit too abrupt. 

For figure 4, there is no point in showing the gel images as these represent end point PCR. As these can not (and was not) be used for quantification I suggest deleting them as they do not provide any relevant information. 

Other than that, the manuscript is of excellent quality and merits publication. Well done! 

Author Response

Reviewer 2

This is a manuscript that is very nicely written and discussed. The introduction is excellent with just a few minor flaws (such as different fonts, for example on line 48). 

In the result section, I suggest a better transition when going from the human subject to the mice. It feels a bit too abrupt. 

The authors are thankful for your careful work that helped to improve the manuscript.

 The transition between sentences was improved.

For figure 4, there is no point in showing the gel images as these represent end point PCR. As these can not (and was not) be used for quantification I suggest deleting them as they do not provide any relevant information. 

The authors agree and figure was changed

Reviewer 3 Report

The authors have adequately dealt with my comments. I have nothing to add.

Author Response

Revisor 3

The authors are thankful for your careful work that helped to improve the manuscript.

Round 2

Reviewer 1 Report

The authors properly addressed the reviewer's concerns.

This manuscript is a resubmission of an earlier submission. The following is a list of the peer review reports and author responses from that submission.

Round 1

Reviewer 1 Report

Machado-Lima and coworkers investigated the role of RAGE in cholesterol efflux in the presence and absence of advanced glycated albumin in mouse macrophages. To this end, the authors isolated macrophages from the bone marrow of Ager null and wild type mice and also used THP1 cells with a knockdown of AGER to perform cholesterol efflux experiments and mRNA determination of proteins involved in cholesterol efflux. The authors used blood from patients with type 1 and type 2 as well as controls for isolating serum albumin and in addition in vitro chemically glycated human albumin. The authors found that AGER expression was needed for an advanced glycated albumin-mediated reduction of cholesterol efflux. In addition, albumin from diabetic patients lead in Wt and Ager null macrophages to differential changes in the mRNA expression of the Nox4, Scd1, Nfkb1, Jak 2 and Abcg1 genes. The authors conclude that RAGE is needed for the reduction of cholesterol efflux by advanced glycated albumin and this pathway could be a novel therapeutic target for advanced glycated albumin induced atherosclerosis.

Comments

1.     The main problem of this manuscript is the lack of protein expression data. If the authors would test the observed changes in mRNA also on Western blots, the manuscript would gain a lot of impact.

2.     Abstract: AGE should be spelled out when it is used for the first time.

3.     Table 1: The authors should give the normal values of the serum parameters in the legend to the table.

4.     Figure 3 should be complemented by cholesterol efflux measurements in control and knock-down THP-1 cells.

5.     The discussion is rather long.

Author Response

Dear Sindy Li,

The authors of the manuscript entitled “RAGE mediates cholesterol efflux impairment in macrophages caused by human advanced glycated albumin” would like to thank the reviewers for their detailed analysis of the manuscript and proper suggestions in order to improve it. Many comments regarding general format and spelling, as well as proper description of the material and methods and discussion sessions can be easily performed.

In regard to the measurement of protein expression by immunoblot, according to the comments raised by the reviewers 1 and 3, the authors totally agree that it will add a very important piece of information considering the disparity between gene expression and protein levels that has been demonstrated for many genes involved in the macrophage cholesterol efflux pathway. In addition, the measurement of cholesterol efflux rate in Ager knockdown THP-1 cells was also required by the reviewer 1 to endorse findings with RAGE knockout macrophages. Considering that those experiments are time consuming and requires antibodies purchasing in Brazil, the authors kindly ask for a time extension for sending the revised form of the manuscript – August 31st, 2019. 

Sincerely yours,

Marisa Passarelli

Reviewer 2 Report

This is a nicely written article which has a few flaws. First of all the text is overall nice but has some sections that need improvements. For example, in the result section it is difficult to distinguish between the data originating from humans, cells or mouse tissue. A clearer structure is needed to avoid confusing the reader. 

Throughout the text there are a few phrases which need to be edited. For example, on line 92 the authors write "The absence of Ager was able to prevent...". This, and similar, spoken language expressions need to be edited. 

In the discussion the authors write about their data but need to better put it in context in relation to the current literature. 

The materials and methods section needs to be rewritten as it is too 'shallow'. The purpose of this section is to provide enough information for the reader to be able to replicate the described experiments. That is currently not possible. 

Also, the authors do not mention which test they used to test for normal distribution. It is possible that the very few human subjects showed Gaussian arrangement but the test of that needs to be shown. Naturally, the same applies to the in vitro data. 

Author Response

(The authors gave the same response as above.)

Reviewer 3 Report

The work presented by Machado-Lima and colleagues is aimed at evaluating the involvement of RAGE in the alterations of macrophage cholesterol efflux caused by glycated albumin. The main results demonstrated that glycated albumin (derived from DM patients, or "in vitro" glycated albumin) strongly suppress cholesterol efflux from WT macrophages, conversely this effect was completely prevented in Ager-null macrophages. Nox4, Jak2, scd-1 and nfkb1 mRNA expression were shown to be altered upon DM albumin treatment in WT cells. On the contrary, these modulations were abrogated in Ager-null cells. Similar results were obtained in another experimental model (THP-1 cells transfected with scramble siRNA and siRNA-Ager). Taken together, this work demonstrated that inhibition of AGE/RAGE axis may be taken into consideration as putative therapeutic approaches to conteract AGE-induced atherosclerosis.

When evaluated as a whole, the manuscript is well written, the presentation of results is good, as well as their integration within the state of the art in the "discussion" section. However, there are some shortcomings that should be addressed. I think that, with some improvements, this experimental work could be of great relevance, thus I hope to see a revised version of it.

1) The manuscript should be carefully revised since some typographical/grammar errors are present

2) In order to increase the coherence between the two experimental models (macrophages derived from WT and ager-null mice; vs transfected THP-1 cells), nfkb1 evaluation should be performed also in THP-1 experiments.

3) As also stated by the authors, alterations observed in transcripts (mRNA) are not necessarily paralleled by changes in the protein expression. The evaluation of protein expression is mandatory in order to support functional speculations and to reinforce the results presented in this work. For these reasons, the molecular targets evaluated in this work (Nox4, Jak2, scd-1, Abca1, Abcg1 and nfkb1) should be also assessed by Western blot. Notably, the analysis of protein expression will be of particular importance for better defining the contribution of Abca1, whose mRNA expression is sometimes not coherent among the proposed sets of experiments. In addition, mRNA estimation of transcription factors and/or signaling proteins (e.g. Jak2, nfkb1) does not make sense. Indeed, the most reasonable and logic approach to evaluate the involvement of Jak2 and NF-kB is to evaluate their activating phosphorylations through the use of specific phospho-antibodies.

5) I think this work will acquire even more strength if the authors will include experiments showing intracellular lipid accumulation among the different experimental conditions.

4) Data are expressed as the mean ± SEM. The standard error of the mean indicates the uncertainty of how the sample mean represents the population mean. In my opinion, the authors inappropriately report the SEM instead of the Standard Deviation (SD). Since the SEM is always less than the SD, it deceives the reader into underestimating the variability between replicates/individuals within the study sample. Thus, SEM should be substituted by SD.

Author Response

(The authors gave the same response as above.)
